# Neural Multivariate Regression with Multi-Task Learning and Target Preprocessing

**George Andriopoulos**[1*]  **Soyuj Jung Basnet**[3]  **Juan Guevara**[4]  **Bimarsha Adhikari**[1]  **Li Guo**[2]  **Keith Ross**[1]

[1] NYU Abu Dhabi   [2] NYU Shanghai   [3] NYU   [4] MBZUAI

## Abstract

The Unconstrained Feature Model (UFM) enables closed-form approximations for training loss in deep neural networks (DNNs). We use the UFM to motivate testable hypotheses about neural multivariate regression—fundamental to imitation learning, robotics, and reinforcement learning. Specifically, we analyze multi-task versus single-task models and the impact of target preprocessing. The UFM predicts advantages for multi-task models under comparable or stronger regularization and identifies regimes where whitening or normalization reduces training error. Experiments across four robotic and autonomous driving datasets consistently support these qualitative trends. This work illustrates how simplified analytical models can structure empirical investigation by generating theoretical predictions that are then validated through practice.

## 1 Introduction

Neural multivariate regression is fundamental to imitation learning, robotics, and deep reinforcement learning. Despite its importance, design choices—such as training a single multi-task model versus multiple dedicated ones, or whether to preprocess targets—remain largely driven by empirical trial-and-error.

We use the Unconstrained Feature Model (UFM) (Fang et al., 2021; Mixon et al., 2022) to motivate testable hypotheses for these questions. The UFM treats last-layer features as free variables, providing a tractable framework with closed-form solutions. While UFM makes strong simplifying assumptions, examining whether its predictions hold in practice offers a principled way to structure empirical investigation into multivariate regression.

We test two concrete hypotheses:

1. **Multi-task vs. single-task:** A multi-task model matches or outperforms $n$ single-task models under equivalent regularization.
2. **Target preprocessing:** Whitening or normalization reduces training error if the average target variance $\bar{\lambda} < 1$, but increases it otherwise.

We validate these predictions on four datasets: robotic locomotion tasks (Reacher, Swimmer, Hopper) and autonomous driving (CARLA). Our experiments confirm both hypotheses, demonstrating that UFM-derived claims transfer to real networks trained with standard optimizers.

## 2 Related Work

**The Unconstrained Feature Model.** The UFM (Mixon et al., 2022) and the closely related Layer-Peeled Model (Fang et al., 2021) treat last-layer features as free optimization variables, enabling closed-form analysis of training dynamics. Unlike Neural Tangent Kernel theory (Jacot et al., 2018), which operates in the infinite-width regime, the UFM captures nonlinear feature learning and has successfully explained phenomena such as neural collapse in classification (Papyan et al., 2020; Zhu et al., 2021). Recent work has extended UFM to imbalanced data (Thrampoulidis et al., 2022), various loss functions (Zhou et al., 2022a;b), and multivariate regression (Andriopoulos et al., 2024;

2025). We build on this last line of work, using UFM-derived closed forms to generate testable predictions.

**Multi-task learning.** Trade-offs between multi-task and single-task learning have been studied extensively in computer vision (Kendall et al., 2018; Vandenhende et al., 2021) and reinforcement learning (Rusu et al., 2015; Yu et al., 2020), with several works comparing their relative merits (Fifty et al., 2021; Ruder, 2017). Our goal is complementary: we use a simplified analytical model to generate qualitative, testable hypotheses, which we then examine through controlled experiments.

**Target preprocessing.** Whitening and normalization are standard for inputs (LeCun et al., 1998) and intermediate features (Ioffe & Szegedy, 2015; Huang et al., 2018), and have been used to reduce domain shift in transfer learning (Sun & Saenko, 2016). However, the effect of preprocessing regression *targets* has received little theoretical attention. Our analysis provides a precise criterion for when such preprocessing helps.

## 3 THE UFM AS A HYPOTHESIS-GENERATING FRAMEWORK

**Setup.** Consider multivariate regression with $M$ training examples $\{(\mathbf{x}_i, \mathbf{y}_i), i = 1, \ldots, M\}$, inputs $\mathbf{x}_i \in \mathbb{R}^D$, and targets $\mathbf{y}_i \in \mathbb{R}^n$. A typical DNN produces outputs $f_{\theta, \mathbf{W}, \mathbf{b}}(\mathbf{x}) = \mathbf{W}\mathbf{h}_\theta(\mathbf{x}) + \mathbf{b}$, where $\mathbf{h}_\theta(\cdot) : \mathbb{R}^D \to \mathbb{R}^d$ is the feature extractor, $\mathbf{W} \in \mathbb{R}^{n \times d}$ is the final layer, and $\mathbf{b} \in \mathbb{R}^n$ is the bias.

**The UFM.** The UFM (Fang et al., 2021; Mixon et al., 2022) replaces the parameterized feature extractor with free variables $\mathbf{H} := [\mathbf{h}_1 \cdots \mathbf{h}_M] \in \mathbb{R}^{d \times M}$, yielding:

$$\mathcal{L}(\mathbf{H}, \mathbf{W}, \mathbf{b}) := \frac{1}{2M}||\mathbf{W}\mathbf{H} + \mathbf{b}\mathbf{1}_M^T - \mathbf{Y}||_F^2 + \frac{\lambda_{\mathbf{H}}}{2M}||\mathbf{H}||_F^2 + \frac{\lambda_{\mathbf{W}}}{2}||\mathbf{W}||_F^2, \quad (1)$$

where $\mathbf{Y} := [\mathbf{y}_1 \cdots \mathbf{y}_M] \in \mathbb{R}^{n \times M}$. This problem admits closed-form solutions, enabling analytical predictions.

**Closed-form training MSE.** Let $\mathbf{\Sigma}$ be the sample covariance of targets with eigenvalues $\lambda_1 \geq \cdots \geq \lambda_n > 0$. Define the regularization product $c := \lambda_{\mathbf{H}}\lambda_{\mathbf{W}}$. Let $j^* := \max\{j : \lambda_j \geq c\}$ with the convention that $j^* = 0$ when the set in question is the empty set. In experiments, we set $\lambda_{\mathbf{H}} = \lambda_{\mathbf{W}} = \lambda_{WD}$ (weight decay), so $c = \lambda_{WD}^2$.

**Theorem 3.1** (Andriopoulos et al. (2024)). *At any global minimum $(\mathbf{H}^*, \mathbf{W}^*, \mathbf{b}^*)$ of (1):*

$$\text{MSE}(\mathbf{H}^*, \mathbf{W}^*, \mathbf{b}^*) = j^*c + \sum_{i=j^*+1}^{n} \lambda_i. \quad (2)$$

This closed-form expression enables deriving predictions for multi-task vs. single-task models and for target preprocessing effects. We state these predictions here in simplified form as testable hypotheses; the full theorem statements and proofs are deferred to Appendix C–E.

### 3.1 HYPOTHESIS 1: MULTI-TASK VS. SINGLE-TASK MODELS

For $n$ single-task models with regularization $\tilde{c}$, let $\sigma_i^2$ denote the variance of the $i$-th target dimension. Re-order the indices so that $\sigma_1^2 \geq \sigma_2^2 \geq \cdots \geq \sigma_n^2$. Define $k^* := \max\{j : \sigma_j^2 \geq \tilde{c}\}$.

**Theorem 3.2.** *Let* $\text{MSE}(\text{multi}, c)$ *denote the multi-task MSE from (2), and* $\text{MSE}(\text{n-single}, \tilde{c}) = k^*\tilde{c} + \sum_{i=k^*+1}^{n} \sigma_i^2$ *the total single-task MSE.*

*(i) If $c = \tilde{c}$, then* $\text{MSE}(\text{multi}, c) \leq \text{MSE}(\text{n-single}, c)$*, with strict inequality when $\lambda_{\min} < c < \lambda_{\max}$ and $j^* < k^*$.*

*(ii) If $c < \tilde{c}$, then* $\text{MSE}(\text{multi}, c) \leq \text{MSE}(\text{n-single}, \tilde{c})$*, with strict inequality when $c < \min\{\tilde{c}, \lambda_{\max}\}$.*

**Prediction:** Multi-task models achieve lower or equal training MSE compared to single-task models under equivalent regularization, with strict improvement in typical settings where $\lambda_{\min} < c < \lambda_{\max}$.

### 3.2 HYPOTHESIS 2: TARGET WHITENING AND NORMALIZATION

**Theorem 3.3.** *Let $\bar{\lambda} := n^{-1}\sum_{i=1}^{n} \lambda_i$ denote the average variance across target dimensions. Using ZCA whitening (see Appendix E):*

($i$) *For $0 < c < \lambda_{\min}$:* MSE(de-whiten) < MSE(no-whitening) *if and only if $\bar{\lambda} < 1$.*

($ii$) *For $c > 1$:* MSE(de-whiten) $\geq$ MSE(no-whitening).

*Normalization yields identical results when $c < \min\{\lambda_{\min}, \tilde{\lambda}_{\min}\}$ (Theorem E.5).*

**Prediction:** Whitening/normalizing targets reduces training MSE when the average target variance $\bar{\lambda} < 1$, and increases MSE when $\bar{\lambda} > 1$.

## 4 EXPERIMENTAL VALIDATION

**Datasets.** We use four datasets spanning different input modalities and target dimensions (Table 1). Reacher, Swimmer, and Hopper are MuJoCo robotic locomotion tasks (Brockman et al., 2016; Todorov et al., 2012; Towers et al., 2024); CARLA 2D is an autonomous driving dataset (Codevilla et al., 2018). Crucially, the MuJoCo datasets have $\bar{\lambda} < 1$ while CARLA has $\bar{\lambda} \gg 1$, enabling a direct test of Hypothesis 2.

Table 1: **Spectral Properties of Datasets.** $\lambda_{\min}$, $\lambda_{\max}$, and $\bar{\lambda}$ denote the minimum, maximum, and average eigenvalues of the target variable's covariance matrix. $\tilde{\lambda}_{\min}$ and $\tilde{\lambda}_{\max}$ represent the minimum and maximum eigenvalues of the target variable's correlation matrix.

| Dataset | $n$ | $\lambda_{\min}$ | $\tilde{\lambda}_{\min}$ | $\lambda_{\max}$ | $\tilde{\lambda}_{\max}$ | $\bar{\lambda}$ |
|---------|-----|---------|---------|---------|---------|---------|
| Reacher | 2 | 0.010 | 0.991 | 0.012 | 1.009 | 0.011 |
| Swimmer | 2 | 0.276 | 0.756 | 0.466 | 1.244 | 0.371 |
| Hopper | 3 | 0.215 | 0.782 | 0.442 | 1.258 | 0.345 |
| CARLA 2D | 2 | 0.024 | 0.996 | 209.097 | 1.004 | 104.561 |

**Models.** For MuJoCo, we use 4-layer MLPs (256 hidden units) following standard practice (Tarasov et al., 2024). For CARLA, we use ResNet18 (He et al., 2016). All models trained with SGD across weight decay values $\lambda_{WD} \in [10^{-5}, 10^{-1}]$. Full experimental details in Appendix A.

### 4.1 RESULTS: HYPOTHESIS 1 (MULTI-TASK VS. SINGLE-TASK)

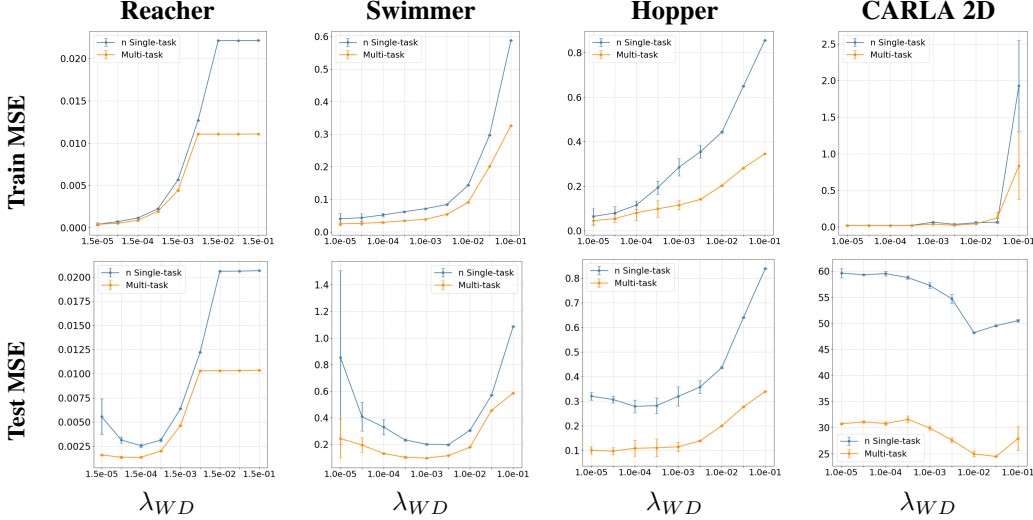

Figure 1: Comparison of the training error (top) and test error (bottom) of a single multi-task model with that of multiple single-task models for different weight decay values after training with the standard parameter-regularized loss function.

Figure 1 shows, regardless of the value of weight decay, that the multi-task model achieves lower training MSE than the sum of single-task models. These empirical results are consistent with

Theorem 3.2, which establishes that under the UFM assumption, multi-task models achieve smaller training MSE when using equivalent or stronger regularization for single-task models. The results are robust to changes in architecture (see Figures 3–5 in Appendix B). Importantly, the advantage extends to test MSE (bottom row), indicating that multi-task learning improves generalization.

## 4.2 RESULTS: HYPOTHESIS 2 (TARGET WHITENING/NORMALIZATION)

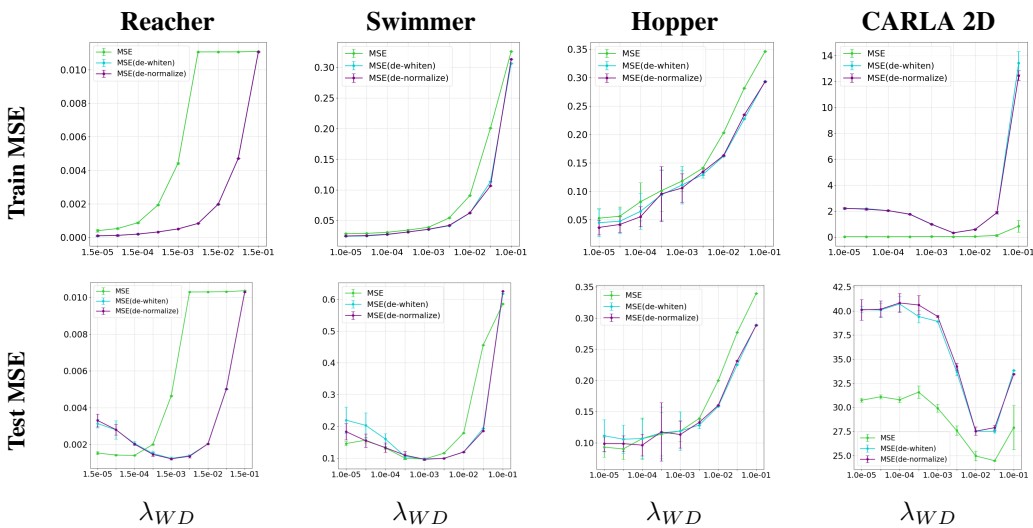

Figure 2: Comparison of the effect that target whitening and normalization have on training error (top) and test error (bottom) for different weight decay values after training with the standard parameter-regularized loss function. The green curve (MSE) records the error after training with the original unprocessed targets.

Figure 2 presents the experimental results, which closely align with the theory. First, as predicted by Theorem E.5, there is minimal difference in training MSE between whitening and normalizing. Second, the impact of whitening/normalizing on training MSE depends on the average eigenvalue $\bar{\lambda}$: whitening/normalizing reduces training MSE when $\bar{\lambda} < 1$ (MuJoCo datasets), but increases training MSE when $\bar{\lambda} > 1$ (CARLA 2D), exactly as Theorem 3.3 predicts. These results are robust to architecture changes (see Figures 6–8 in Appendix B). The trends for test MSE align closely with training MSE.

These results provide practical insights: practitioners should compute $\bar{\lambda}$ before applying target preprocessing. For imitation learning tasks with action targets in small ranges, $\bar{\lambda} < 1$ is common and whitening helps. For tasks with large-magnitude targets (e.g., speed in CARLA spans $[0, 85]$), whitening can hurt.

## 5 CONCLUSION

This paper shows that the Unconstrained Feature Model (UFM), despite its simplifying assumptions, yields qualitative predictions that hold in practice. We derive closed-form results addressing two fundamental questions in neural multivariate regression—multi-task versus single-task models and target preprocessing—and validate these predictions across four datasets spanning robotic control and autonomous driving.

The broader significance lies in the methodology: minimal analytical models can generate precise, conditional predictions (e.g., whitening helps iff $\bar{\lambda} < 1$) that transfer to real networks trained with standard optimizers. This suggests that tractable theory, even when approximate, can guide architectural and preprocessing choices. Overall, this study illustrates how theoretical abstractions, paired with targeted experiments, yield testable insights into learning dynamics.

ACKNOWLEDGMENTS

This work was supported by the NYU Abu Dhabi Center for Artificial Intelligence and Robotics, funded by Tamkeen under the Research Institute Award CG010; the NYU Abu Dhabi Center for Interdisciplinary Data Science and AI; and the Shanghai Frontiers Science Center of Artificial Intelligence and Deep Learning at NYU Shanghai.

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

# A  EXPERIMENTAL DETAILS

## A.1  MuJoCo

The datasets Reacher and Swimmer are sourced from an open-source repository (Gallouédec et al., 2024) and consist of expert data generated by a policy trained using Proximal Policy Optimization (PPO) (Schulman et al., 2017). For Hopper, the dataset is part of the D4RL benchmark (Fu et al., 2020), which is widely recognized in offline reinforcement learning research. Table 2 provides a summary of all model hyper-parameters and experimental configurations. In all experiments, the models are trained until their weights converge.

Table 2: Hyper-parameter settings for experiments with weight decay on MuJoCo datasets.

|  | Hyper-parameter | Value |
|---|---|---|
| Model Architecture | Number of hidden layers | 3 |
|  | Hidden layer dimension | 256 |
|  | Activation function | ReLU |
|  | Number of linear projection layer ($\mathbf{W}$) | 1 |
| Training | Epochs | 2e5, Reacher |
|  |  | 2e5, Swimmer |
|  |  | 4e4, Hopper |
|  | Batch size | 256 |
|  | Optimizer | SGD |
|  | Learning rate | 1e-2 |
|  | Seeds | 0, 1, 2 |
|  | Compute resources | NVIDIA A100 8358 80GB |
|  | Number of compute workers | 4 |
|  | Requested compute memory | 16 GB |
|  | Approximate average execution time | 5 hours |

**MuJoCo environment descriptions**. We utilize expert data from previous work (Gallouédec et al., 2024; Fu et al., 2020) for the Reacher, Swimmer, and Hopper environments. The Reacher environment features a two-jointed robotic arm, where the objective is to control the arm's tip to reach a randomly placed target in a 2D plane. The Swimmer environment consists of a three-segment robot connected by two rotors, designed to propel itself forward as quickly as possible. Similarly, the Hopper environment is a single-legged robot with four connected body parts, aiming to hop forward efficiently.

**Low data regime**. Training neural networks with expert state-action data using regularized regression is commonly known as *imitation learning*. Following standard practices for MuJoCo environments (Tarasov et al., 2024), we employ relatively small multi-layer perceptron (MLP) architectures. Since the goal in imitation learning is to achieve strong performance with minimal expert data, we train models using only a fraction of the available datasets. Specifically, we use 20 expert demonstrations (episodes) for Reacher, 1 for Swimmer, and 10 for Hopper, translating to datasets of 1,000, 1,000, and 10,000 samples respectively.

## A.2  CARLA

The CARLA dataset is created by capturing the vehicle's surroundings through automotive cameras while a human driver controls the vehicle in a simulated urban environment (Codevilla et al., 2018). The recorded images represent the vehicle's states $\mathbf{x}_i$, and the expert driver's control inputs, including speed and steering angles, are treated as the actions $\mathbf{y}_i \in [0, 85] \times [-1, 1]$ in the dataset.

For feature extraction from images, we use ResNet-18 (He et al., 2016) as the backbone model. Since a large number of images is required to train a robust feature extractor from visual inputs (He et al., 2016; Sun et al., 2017), the entire dataset is utilized for training. The experimental setup for CARLA is detailed in Table 3.

Table 3: Hyper-parameters of ResNet for CARLA dataset.

|  | **Hyper-parameter** | **Value** |
|---|---|---|
| Architecture | Backbone of hidden layers | ResNet18 |
|  | Last layer hidden dim | 512 |
| Training | Epochs | 100 |
|  | Batch size | 512 |
|  | Optimizer | SGD |
|  | Momentum | 0.9 |
|  | Learning rate | 0.001 |
|  | Seeds | 0, 1 |
|  | Compute resources | NVIDIA A100 8358 80GB |
|  | Number of compute workers | 8 |
|  | Requested compute memory | 200 GB |
|  | Approximate average execution time | 42 hours |

### A.3 INTER-TASK CORRELATIONS

Our chosen datasets exhibit a range of task correlations. Table 4 includes the Pearson correlation coefficient between the $i$-th and $j$-th target components for $i \neq j$. From Table 4, we observe that the target components in CARLA 2D and Reacher are nearly uncorrelated, whereas those in Hopper and Swimmer exhibit stronger correlations. This demonstrates that multi-task learning's advantages in our experiments are not solely attributable to high-correlation scenarios.

Table 4: Overview of datasets employed in our analysis.

| **Dataset** | **Data Size** | **Input Type** | **Target Dimension** $n$ | **Target Correlation** |
|---|---|---|---|---|
| Swimmer | 1K | raw state | 2 | -0.244 |
| Reacher | 10K | raw state | 2 | -0.00933 |
| Hopper | 10K | raw state | 3 | [-0.215, -0.090, 0.059] |
| CARLA 2D | 600K | RGB image | 2 | -0.0055 |

## B ADDITIONAL EXPERIMENTAL RESULTS

### B.1 ROBUSTNESS ACROSS ARCHITECTURES

To explore how network capacity impacts our findings, we repeated experiments across multiple architectures, covering a broad range of widths and depths. The results in Figures 3–5 demonstrate that, regardless of architecture, multi-task models outperform single-task models (confirming Hypothesis 1). Figures 6–8 confirm Hypothesis 2 across architectures.

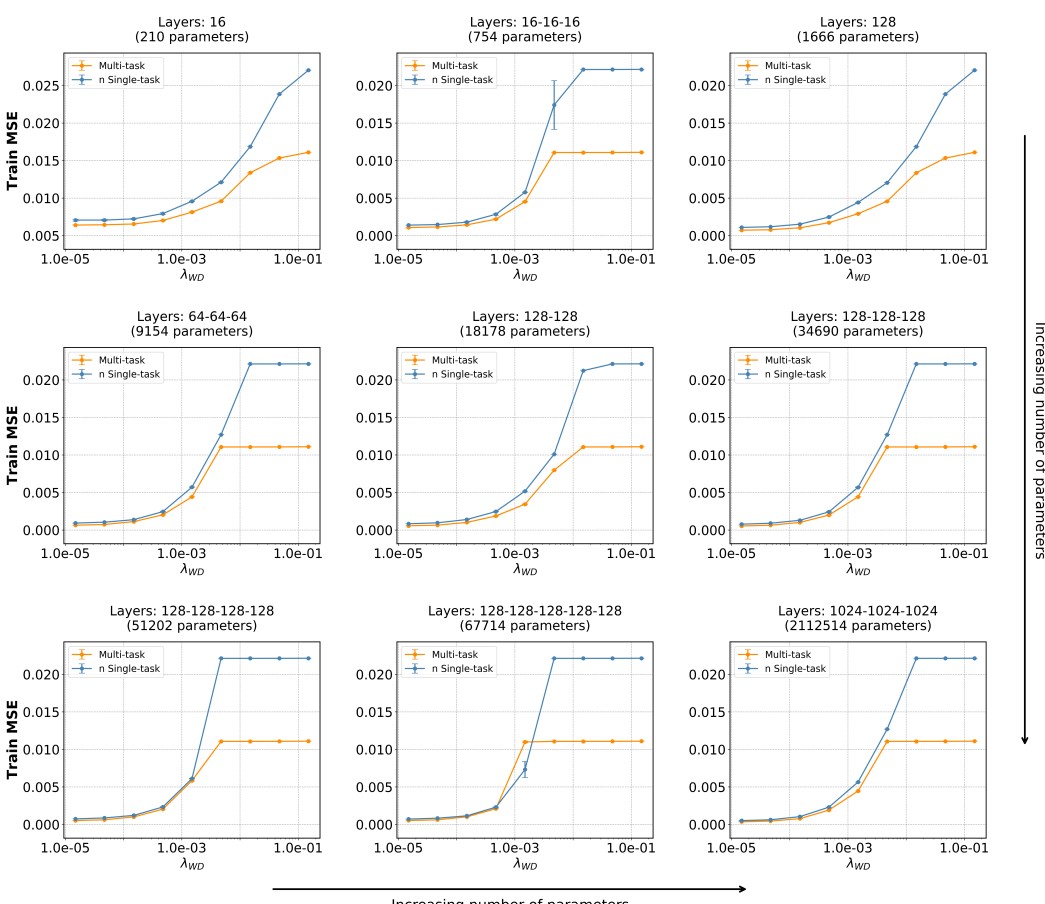

Figure 3: **Effect of network architecture on multi-task vs. single-task training: Reacher.** Comparison of the training error of a single multi-task model with that of multiple single-task models for different weight decay values after training with the standard parameter-regularized loss function across different architectures. The architectures are denoted by their layer sizes (input and output layers omitted for simplicity). The number of parameters increases from left to right and from top to bottom.

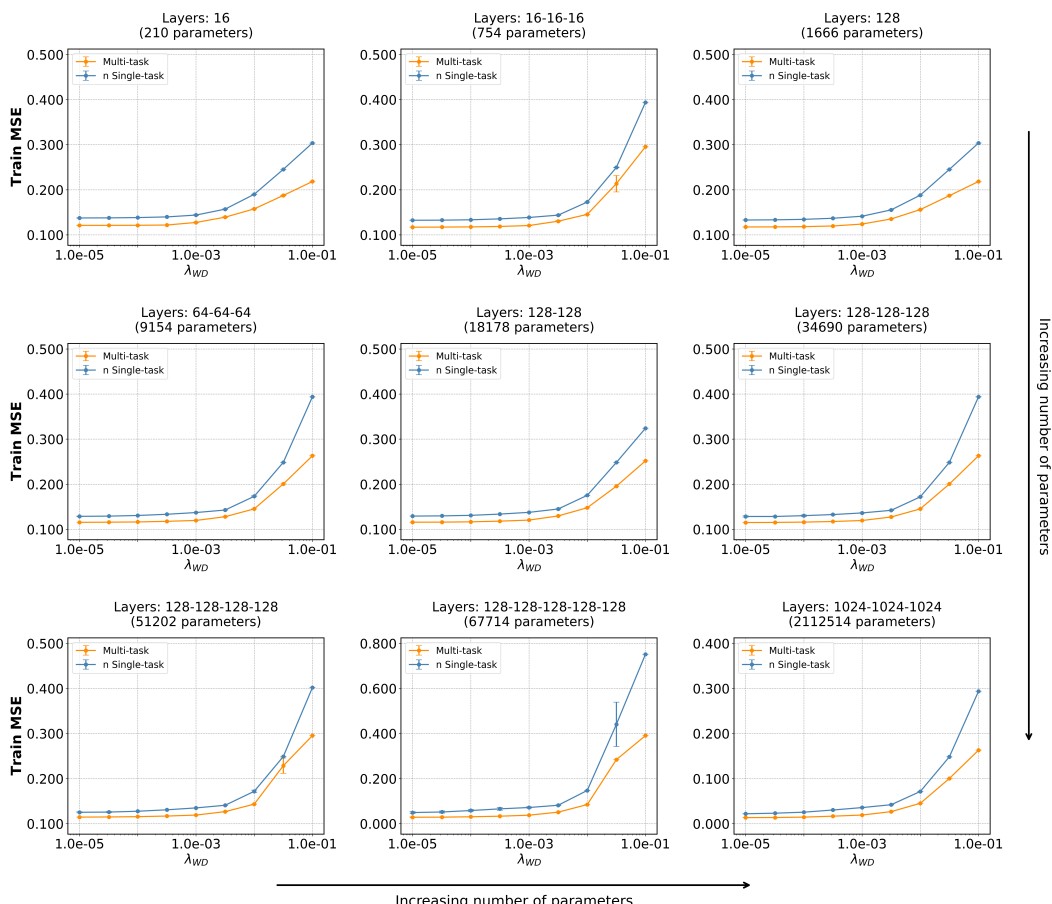

Figure 4: **Effect of network architecture on multi-task vs. single-task training: Swimmer.** Comparison of the training error of a single multi-task model with that of multiple single-task models for different weight decay values after training with the standard parameter-regularized loss function across different architectures. The architectures are denoted by their layer sizes (input and output layers omitted for simplicity). The number of parameters increases from left to right and from top to bottom.

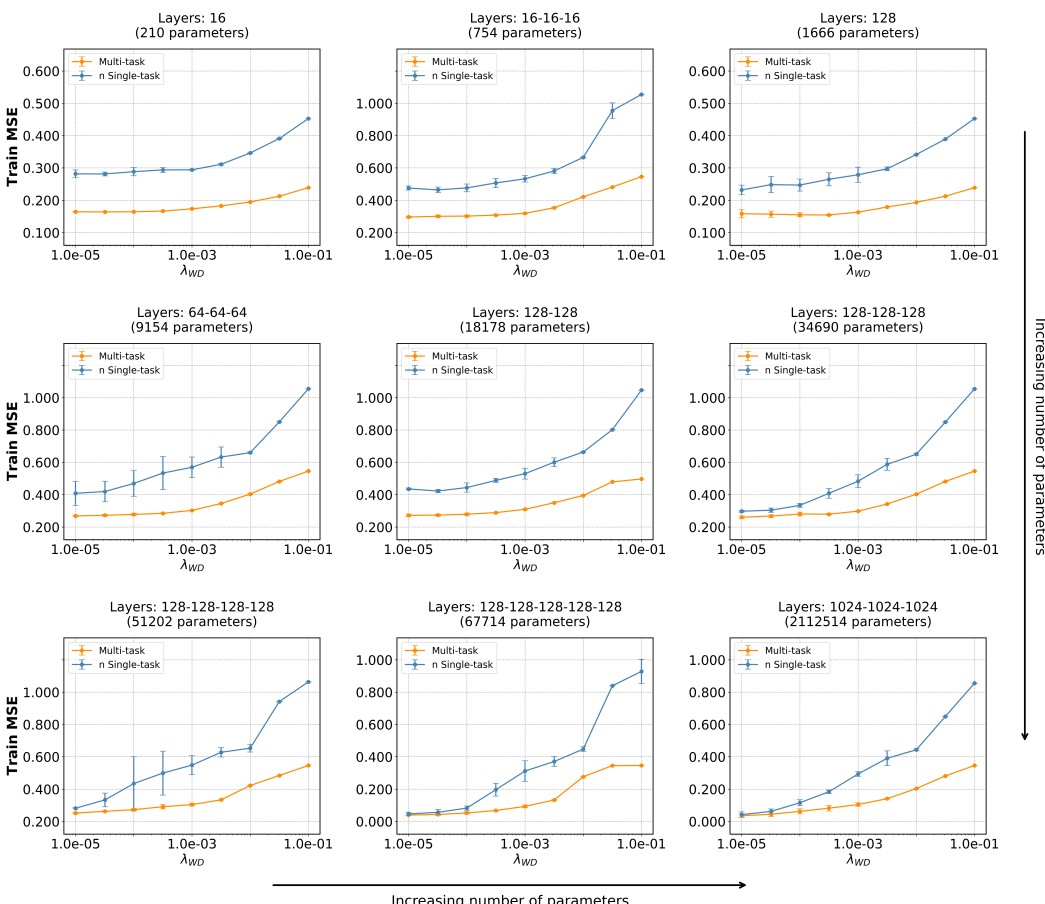

Figure 5: **Effect of network architecture on multi-task vs. single-task training: Hopper.** Comparison of the training error of a single multi-task model with that of multiple single-task models for different weight decay values after training with the standard parameter-regularized loss function across different architectures. The architectures are denoted by their layer sizes (input and output layers omitted for simplicity). The number of parameters increases from left to right and from top to bottom.

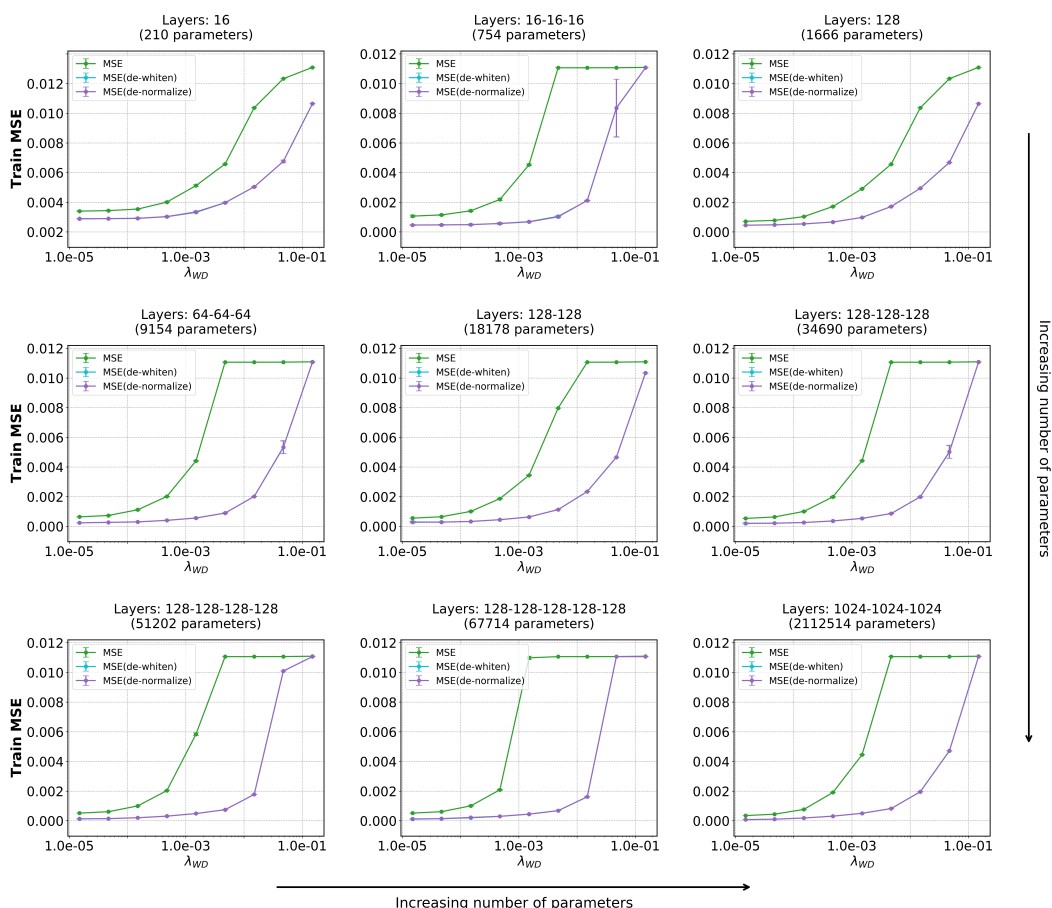

Figure 6: **Whitening vs. normalization vs. raw targets: Reacher.** Comparison of the effect that target whitening and normalization have on training error for different weight decay values after training with the standard parameter-regularized loss function across different architectures. The green curve (MSE) records the training error after training with the original unprocessed targets. The architectures are denoted by their layer sizes (input and output layers omitted for simplicity). The number of parameters increases from left to right and from top to bottom.

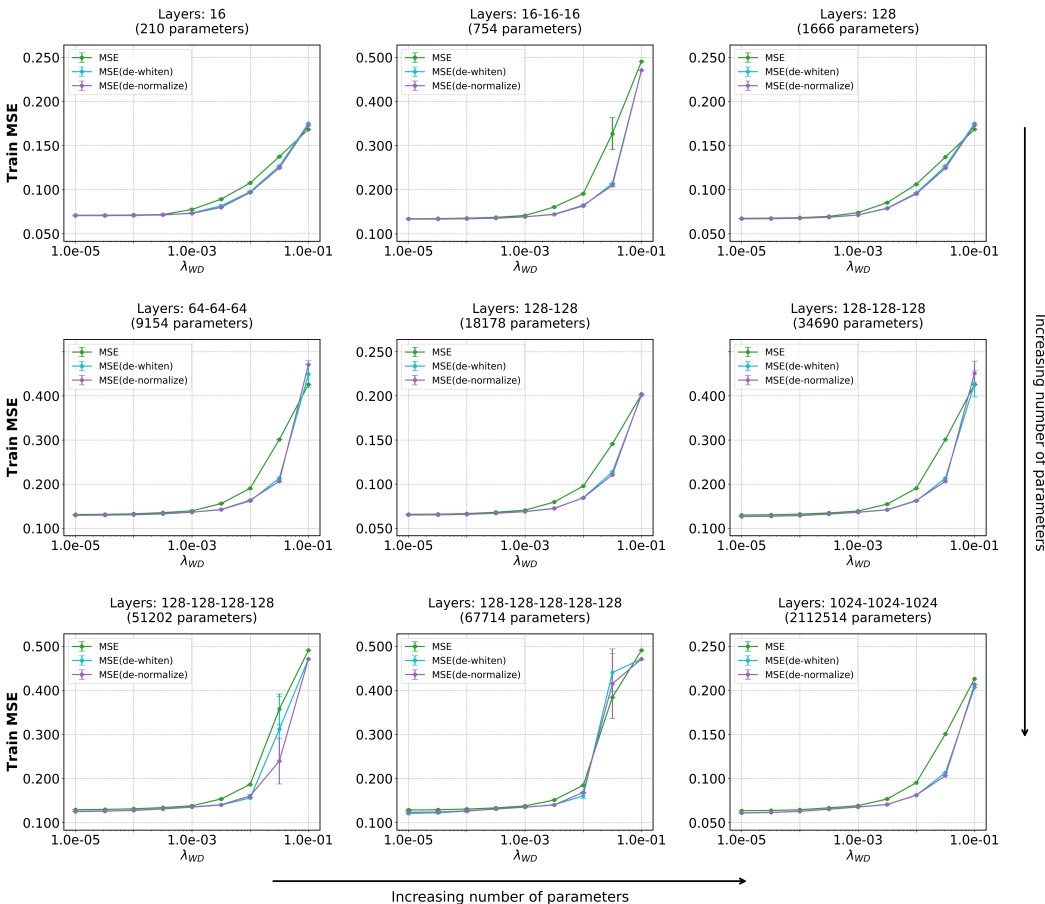

Figure 7: **Whitening vs. normalization vs. raw targets: Swimmer.** Comparison of the effect that target whitening and normalization have on training error for different weight decay values after training with the standard parameter-regularized loss function across different architectures. The green curve (MSE) records the training error after training with the original unprocessed targets. The architectures are denoted by their layer sizes (input and output layers omitted for simplicity). The number of parameters increases from left to right and from top to bottom.

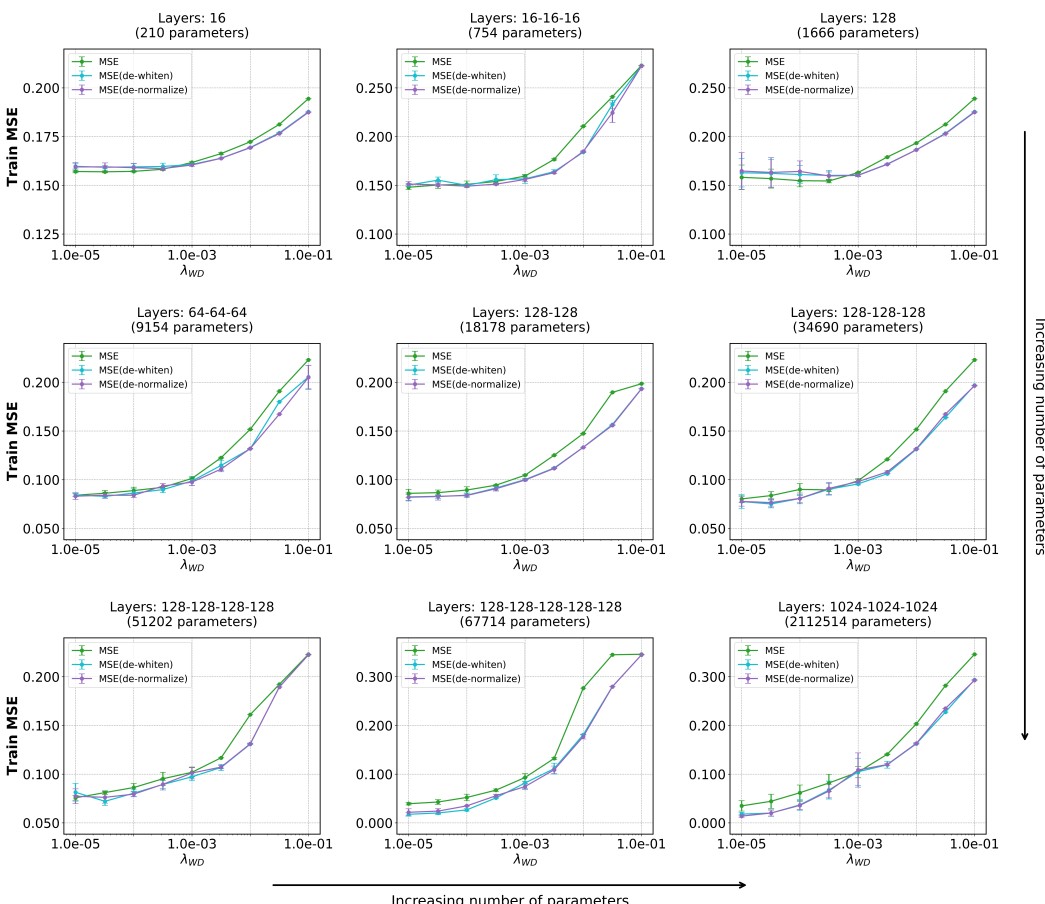

Figure 8: **Whitening vs. normalization vs. raw targets: Hopper.** Comparison of the effect that target whitening and normalization have on training error for different weight decay values after training with the standard parameter-regularized loss function across different architectures. The green curve (MSE) records the training error after training with the original unprocessed targets. The architectures are denoted by their layer sizes (input and output layers omitted for simplicity). The number of parameters increases from left to right and from top to bottom.

## B.2 TEST MSE RESULTS

For completeness, we report test MSE, which shows consistent trends with training MSE:

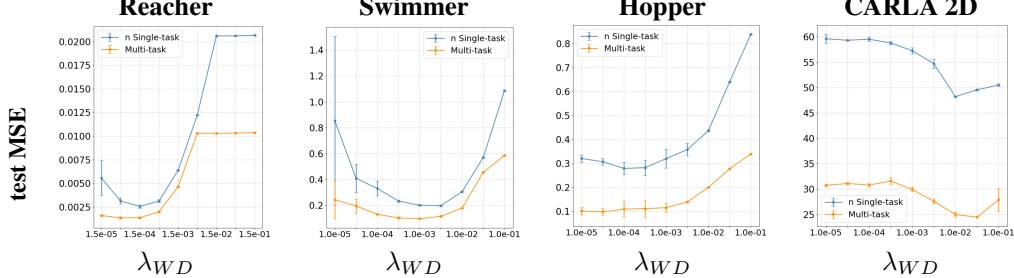

Figure 9: **Test MSE: multi-task vs. single-task models.** Multi-task learning consistently achieves lower test MSE than single-task learning across all datasets and weight decay values, supporting the intuition that multi-task regression improves generalization by learning shared patterns through weights and representation sharing in a single model.

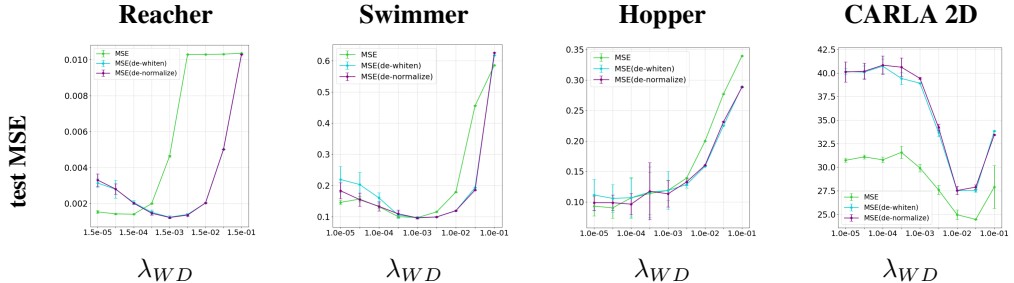

Figure 10: **Test MSE: whitening and normalization effects.** The difference between whitening and normalization is minor. When the average eigenvalue is $< 1$ (MuJoCo datasets), whitening/normalization generally reduces test MSE. When the average eigenvalue is $> 1$ (CARLA 2D), whitening/normalization increases test MSE, reflecting the same trend observed for training MSE.

## C PROOF OF THEOREM 3.1

*Proof of Theorem 3.1.* Let $\tilde{\mathbf{Y}} = \mathbf{Y} - \bar{\mathbf{Y}} = U\tilde{\Sigma}V^T$ denote the compact SVD of $\tilde{\mathbf{Y}}$, where $U \in \mathbb{R}^{n \times n}$ is orthogonal, $V \in \mathbb{R}^{M \times n}$ is semi-orthogonal, and $\tilde{\mathbf{\Sigma}} \in \mathbb{R}^{n \times n}$ is diagonal containing the singular values $\eta_1 \geq \eta_2 \geq \cdots \geq \eta_n > 0$. Let $(\mathbf{H}^*, \mathbf{W}^*, \mathbf{b}^*)$ be a global minimum of (1). By Lemma B.1(Zhou et al., 2022a), the associated MSE is

$$\text{MSE}(\mathbf{H}^*, \mathbf{W}^*, \mathbf{b}^*) = \frac{1}{M} \sum_{i=1}^{n} ([\eta_i - \sqrt{M c}]_+ - \eta_i)^2. \tag{3}$$

Furthermore, using the SVD of $\tilde{\mathbf{Y}} = U\tilde{\Sigma}V^T$,

$$\mathbf{\Sigma} = \frac{\tilde{\mathbf{Y}}\tilde{\mathbf{Y}}^T}{M} = U\frac{\tilde{\mathbf{\Sigma}}}{\sqrt{M}}V^T V \frac{\tilde{\mathbf{\Sigma}}}{\sqrt{M}}U^T = U\left[\frac{\tilde{\mathbf{\Sigma}}}{\sqrt{M}}\right]^2 U^T,$$

from which we have $\mathbf{\Sigma}^{1/2} = U\frac{\tilde{\mathbf{\Sigma}}}{\sqrt{M}}U^T$. This further yields

$$\sqrt{M}[\mathbf{\Sigma}^{1/2} - \sqrt{c}\mathbf{I}_n] = U\tilde{\mathbf{\Sigma}}U^T - U\sqrt{Mc}\mathbf{I}_n U^T$$

Since $U^T = U^{-1}$,

$$\sqrt{M}[\mathbf{\Sigma}^{1/2} - \sqrt{c}\mathbf{I}_n] = U\left[\tilde{\mathbf{\Sigma}} - \sqrt{Mc}\mathbf{I}_n\right]U^{-1}, \tag{4}$$

which implies that the matrices $\sqrt{M}[\boldsymbol{\Sigma}^{1/2} - \sqrt{c}\mathbf{I}_n]$ and $\tilde{\boldsymbol{\Sigma}} - \sqrt{Mc}\mathbf{I}_n$ are similar. As a result, they have the same eigenvalues. The $n \times n$ matrix on the left-hand side of (4) has eigenvalues given by $\sqrt{M\lambda_i} - \sqrt{Mc}$, $i = 1, ..., n$, where $\lambda_i$ is the $i$-th eigenvalue of $\boldsymbol{\Sigma}$, whereas the $n \times n$ matrix on the right-hand side of (4) has eigenvalues $\eta_i - \sqrt{Mc}$, $i = 1, ..., n$. Since the eigenvalues in these two sets are both arranged in descending order, we have

$$\sqrt{\lambda_i} = \frac{\eta_i}{\sqrt{M}}, \qquad \text{for all } i = 1, ..., n. \tag{5}$$

Correspondingly, by (3) and (5) we obtain

$$\begin{aligned}
\text{MSE}(\mathbf{H}^*, \mathbf{W}^*, \mathbf{b}^*) &= \frac{1}{M} \sum_{i=1}^{n} ([\eta_i - \sqrt{Mc}]_+ - \eta_i)^2 \\
&= \frac{1}{M} \sum_{i=1}^{n} ([\sqrt{M\lambda_i} - \sqrt{Mc}]_+ - \sqrt{M\lambda_i})^2 \\
&= j^*c + \sum_{i=j^*+1}^{n} \lambda_i
\end{aligned}$$

as desired. When $n = 1$, $\boldsymbol{\Sigma}$ is simply the scalar $\sigma^2$, which together with the equation above completes the proof. $\qquad\square$

Next, we give a simple upper bound of the form (6) as a corollary. The bound is explicitly given as the minimum of the MSEs when $c < \lambda_{\min}$ ($j^* = n$) and $c > \lambda_{\max}$ ($j^* = 0$) respectively.

**Corollary C.1.**

$$\text{MSE}(\mathbf{H}, \mathbf{W}, \mathbf{b}) \leq \min \left\{ nc, \sum_{i=1}^{n} \lambda_i \right\}. \tag{6}$$

*Proof.* Note that since $c \leq \lambda_i$, for all $i \leq j^*$, and $c > \lambda_i$, for all $i > j^*$, we have that

$$\text{MSE}(\mathbf{H}, \mathbf{W}, \mathbf{b}) = j^*c + \sum_{i=j^*+1}^{n} \lambda_i < j^*c + \sum_{i=j^*+1}^{n} c = nc,$$

$$\text{MSE}(\mathbf{H}^*, \mathbf{W}^*, \mathbf{b}^*) = j^*c + \sum_{i=j^*+1}^{n} \lambda_i = \sum_{i=1}^{j^*} c + \sum_{i=j^*+1}^{n} \lambda_i \leq \sum_{i=1}^{j^*} \lambda_i + \sum_{i=j^*+1}^{n} \lambda_i = \sum_{i=1}^{n} \lambda_i.$$

The desired result readily follows. $\qquad\square$

## D  PROOF OF THEOREM 3.2

*Proof of Theorem 3.2.* We begin by using the Schur-Horn theorem to establish some relations between the eigenvalues $\lambda_1, \ldots, \lambda_n$ of $\boldsymbol{\Sigma}$ and the diagonal elements (variances) $\sigma_1^2, \ldots, \sigma_n^2$ of $\boldsymbol{\Sigma}$. Recall that both $\lambda_1, \ldots, \lambda_n$ and $\sigma_1^2, \ldots, \sigma_n^2$ are arranged in descending order. By the Schur-Horn theorem, the vector containing the diagonal elements of $\boldsymbol{\Sigma}$ is majorized by the vector that contains the ordered eigenvalues of $\boldsymbol{\Sigma}$, i.e.,

$$\sum_{i=1}^{k} \lambda_i \geq \sum_{i=1}^{k} \sigma_i^2, \qquad \text{for all } k = 1, \ldots, n-1, \tag{7}$$

$$\sum_{i=1}^{n} \lambda_i = \sum_{i=1}^{n} \sigma_i^2. \tag{8}$$

From (7) and (8) we have

$$\lambda_1 \geq \sigma_1^2 \geq \sigma_2^2 \geq \cdots \geq \sigma_n^2 \geq \lambda_n. \tag{9}$$

and also the following inequalities for the tail partial sums:

$$\sum_{i=k}^{n} \lambda_i \le \sum_{i=k}^{n} \sigma_i^2, \qquad \text{for all } k = 1, \dots, n. \tag{10}$$

Fix $0 < c < \tilde{c}$. We will consider 6 cases:

**Case I:** Suppose $c < \tilde{c} < \lambda_n$. By (9), we have that $\tilde{c} < \sigma_n^2$. Thus, by Theorem 3.1 and Corollary D.2, the two MSEs are given by

$$nc = \text{MSE}(\text{multi}, c) < \text{MSE}(\text{n-single}, \tilde{c}) = n\tilde{c}.$$

Observe that when $c = \tilde{c}$, the two MSEs are equal to $nc$.

**Case II:** Suppose $\tilde{c} \ge c > \lambda_1$. By (9), we have that $\tilde{c} > \sigma_1^2$. Thus, by Theorem 3.1 and Corollary D.2, the two MSEs are given by

$$\text{MSE}(\text{multi}, c) = \sum_{i=1}^{n} \lambda_i = \sum_{i=1}^{n} \sigma_i^2 = \text{MSE}(\text{n-single}, \tilde{c}).$$

**Case III:** Suppose $c < \lambda_n, \tilde{c} > \lambda_1$. From Theorem 3.1 and Corollary D.2, the difference of the two MSEs is given by

$$\text{MSE}(\text{multi}, c) - \text{MSE}(\text{n-single}, \tilde{c}) = nc - \sum_{i=1}^{n} \lambda_i = \sum_{i=1}^{n}(c - \lambda_i) < 0,$$

since $c < \lambda_n \le \lambda_i$, for all $i = 1, ..., n$.

**Case IV:** Suppose $\lambda_n < c < \lambda_1 < \tilde{c}$. From Theorem 3.1 and Corollary D.2, the difference of the two MSEs is given by

$$\text{MSE}(\text{multi}, c) - \text{MSE}(\text{n-single}, \tilde{c}) = j^*c - \sum_{i=1}^{j^*} \lambda_i = \sum_{i=1}^{j^*}(c - \lambda_i) < 0,$$

since $c < \lambda_1$ and $c \le \lambda_i$, for all $i \le j^*$.

**Case V:** Suppose $c < \lambda_n < \tilde{c} < \lambda_1$. From Theorem 3.1 and Corollary D.2, the difference of the two MSEs is given by

$$
\begin{aligned}
\text{MSE}(\text{multi}, c) - \text{MSE}(\text{n-single}, \tilde{c}) &= nc - k^*\tilde{c} - \sum_{i=k^*+1}^{n} \sigma_i^2 \\
&= \sum_{i=k^*+1}^{n}(c - \sigma_i^2) + k^*(c - \tilde{c}) \\
&< \sum_{i=k^*+1}^{n}(\lambda_i - \sigma_i^2) + k^*(c - \tilde{c}) \\
&= \sum_{i=1}^{k^*}(\sigma_i^2 - \lambda_i) + k^*(c - \tilde{c}) < 0,
\end{aligned}
$$

where the first inequality holds since $c < \lambda_n \le \lambda_i$, for all $i = 1, ..., n$, and the last inequality is due to (7) and since $c < \tilde{c}$.

**Case VI:** Suppose $\lambda_n < c \le \tilde{c} < \lambda_1$. Recall that $j^* := \max\{j : \lambda_j \ge c\}$ and $k^* := \max\{j : \sigma_j^2 \ge \tilde{c}\}$. We will consider three subcases.

**VIa):** Suppose $j^* < k^*$. From Theorem 3.1 and Corollary D.2, the difference of the two MSEs is given by

$$\text{MSE}(\text{multi}, c) - \text{MSE}(\text{n-single}, \tilde{c})$$

$$= \sum_{i=j^*+1}^{n} \lambda_i - \sum_{i=k^*+1}^{n} \sigma_i^2 - (k^* - j^*)c + k^*(c - \tilde{c})$$

$$= \sum_{i=j^*+1}^{k^*} \lambda_i + \sum_{i=k^*+1}^{n} \lambda_i - \sum_{i=k^*+1}^{n} \sigma_i^2 - (k^* - j^*)c + k^*(c - \tilde{c})$$

$$= \sum_{i=k^*+1}^{n} (\lambda_i - \sigma_i^2) + \sum_{i=j^*+1}^{k^*} (\lambda_i - c) + k^*(c - \tilde{c}) < 0. \tag{11}$$

The first sum in (11) is non-positive by (10). The second sum is strictly negative since $\lambda_i < c$, for all $i > j^*$. Therefore, $\text{MSE}(\text{multi}, c) < \text{MSE}(\text{n-single}, \tilde{c})$.

**VIb):** Suppose $k^* \leq j^*$. From Theorem 3.1 and Corollary D.2, the difference of the two MSEs is given by

$$\text{MSE}(\text{multi}, c) - \text{MSE}(\text{n-single}, \tilde{c})$$

$$= \sum_{i=k^*+1}^{j^*} (c - \sigma_i^2) + \sum_{i=j^*+1}^{n} (\lambda_i - \sigma_i^2) + k^*(c - \tilde{c})$$

$$= \sum_{i=k^*+1}^{j^*} (c - \sigma_i^2) + \sum_{i=1}^{j^*} (\sigma_i^2 - \lambda_i) + k^*(c - \tilde{c})$$

$$= \sum_{i=k^*+1}^{j^*} (c - \sigma_i^2) + \sum_{i=1}^{k^*} (\sigma_i^2 - \lambda_i) + \sum_{i=k^*+1}^{j^*} (\sigma_i^2 - \lambda_i) + k^*(c - \tilde{c})$$

$$\leq \sum_{i=k^*+1}^{j^*} (\lambda_i - \sigma_i^2) + \sum_{i=1}^{k^*} (\sigma_i^2 - \lambda_i) + \sum_{i=k^*+1}^{j^*} (\sigma_i^2 - \lambda_i) + k^*(c - \tilde{c}) \tag{12}$$

$$= \sum_{i=1}^{k^*} (\sigma_i^2 - \lambda_i) + k^*(c - \tilde{c}) \leq 0, \tag{13}$$

where the inequality in (12) holds due to the fact that $c \leq \lambda_i$, for all $i \leq j^*$, and the inequality (13) is due to (7). Therefore, $\text{MSE}(\text{multi}, c) \leq \text{MSE}(\text{n-single}, \tilde{c})$. $\qquad\square$

*Remark* D.1. *Applicability without L2-regularization*: Our analysis relies on L2-regularization to yield non-trivial closed-form results, that is the UFM-approximation of training MSE in Theorem 3.1 holds when the UFM-regularization constant $c > 0$. If $c = 0$, it is easy to see that, for any $n \times d$ matrix $\mathbf{W}$ with full-rank $n$, considering the set of $\mathbf{H}$ that satisfy $||\mathbf{WH} - \mathbf{Y}||_F^2 = 0$, gives:

$$\mathbf{H} = \mathbf{W}^+ \mathbf{Y} + (\mathbf{I}_d - \mathbf{W}^+ \mathbf{W})\mathbf{Z},$$

where $\mathbf{W}^+$ is the pseudoinverse of $\mathbf{W}$ and $\mathbf{Z}$ is any $d \times M$ matrix, and this is the well-known solution of the standard least squares problem. Thus, comparing multi-task ($c = 0$) with single-tasks ($\tilde{c} = 0$) is trivial as both training MSEs are identical and equal to zero. Such a UFM-inspired approximation agrees with classical notions of overfitting.

**Corollary D.2.** *The total MSE across the $n$ single-task problems is given by*

$$\text{MSE}(\textit{n-single}, \tilde{c}) = k^* \tilde{c} + \sum_{i=k^*+1}^{n} \sigma_i^2. \tag{14}$$

## E   PROOFS OF THEOREMS ON WHITENING AND NORMALIZATION

Before delving into the proofs of Theorems E.3 and 3.3, we provide a brief introduction to the motivation and key concepts relevant to whitening.

In statistical analysis, whitening (or sphering) refers to a common pre-processing step to transform random variables to orthogonality. A whitening transformation (or sphering transformation) is a linear transformation that converts a random vector with a known covariance matrix into a new random vector of the same dimension and with covariance matrix given by the identity matrix. Orthogonality among random vectors greatly simplifies multivariate data analysis both from a computational as well as from a statistical standpoint. Whitening is employed mostly in pre-processing but is also part of modeling, see for instance (Hao et al., 2015; Zuber & Strimmer, 2009).

Due to rotational freedom there are infinitely many whitening transformations. All produce orthogonal but different sphered random variables. To understand differences between whitening transformations, and to select an optimal whitening procedure for a particular situation, the work of (Kessy et al., 2018) provided an overview of the underlying theory and discussed several natural whitening procedures. For example, they identified PCA whitening as the unique procedure that maximizes the compression of all components of the unprocessed vector in each component of the sphered vector. Of particular interest for our work is *ZCA whitening*, ZCA standing for zero-phase component analysis. Rather than dimensionality reduction and data compression, ZCA whitening is useful for retaining maximal similarity between the unprocessed and the transformed variables.

**Definition E.1.** ZCA whitening employs the sphering matrix $\mathbf{W}^{ZCA} = \mathbf{\Sigma}^{-1/2}$, i.e.,

$$\mathbf{Y}^{ZCA} := \mathbf{\Sigma}^{-1/2}(\mathbf{Y} - \bar{\mathbf{Y}}).$$

We make the following observations:

1. Clearly, writing $\mathbf{W}^{ZCA} = \mathbf{Q}_1 \mathbf{\Sigma}^{-1/2}$, where $\mathbf{Q}_1$ is an orthogonal matrix, $\mathbf{W}^{ZCA}$ satisfies

   $$\mathbf{W}^{ZCA}\mathbf{\Sigma}(\mathbf{W}^{ZCA})^T = \mathbf{I}_n,$$

   thus leading to new (of the infinitely many) whitening transformations. In fact, with $\mathbf{Q}_1 = \mathbf{I}_n$, ZCA whitening is the unique sphering method with a symmetric whitening matrix.

2. Breaking the rotational invariance by investigating the cross-covariance between unprocessed (centered) and sphered targets is key to identifying the optimal whitening transformations. The sample *cross-covariance* between $\mathbf{Y}^{ZCA}$ and $\mathbf{Y}$ is given by

   $$\mathbf{\Phi} := \frac{\mathbf{Y}^{ZCA}(\mathbf{Y} - \bar{\mathbf{Y}})^T}{M} = \mathbf{W}^{ZCA}\frac{(\mathbf{Y} - \bar{\mathbf{Y}})(\mathbf{Y} - \bar{\mathbf{Y}})^T}{M} = \mathbf{W}^{ZCA}\mathbf{\Sigma} = \mathbf{Q}_1\mathbf{\Sigma}^{1/2}.$$

   Note that $\mathbf{\Phi}$ is in general not symmetric, unless $\mathbf{Q}_1 = \mathbf{I}_n$. Using targets and their (ZCA)-whitened counterparts, the least squares objective is minimized when the trace of the cross-covariance is maximized, e.g., (Kessy et al., 2018)[eq. (13) and (14)],

   $$\frac{1}{M}||\mathbf{Y}^{ZCA} - (\mathbf{Y} - \bar{\mathbf{Y}})||_F^2 = n - 2\mathrm{tr}(\mathbf{\Phi}) + \sum_{i=1}^n \sigma_i^2 = n - 2\mathrm{tr}(\mathbf{Q}_1\mathbf{\Sigma}^{1/2}) + \sum_{i=1}^n \sigma_i^2.$$

   It can be shown that the minimum of the latter is attained at $\mathbf{Q}_1 = \mathbf{I}_n$. Therefore, not only ZCA whitening is the unique sphering method with a symmetric whitening matrix. It is also the optimal whitening approach identified by evaluating the objective function of the total squared distance between the unprocessed and the whitened targets, computed from the cross-covariance $\mathbf{\Phi}$.

To summarize, ZCA whitening is the unique procedure used with the aim of making the transformed targets as similar as possible to the unprocessed targets, which is appealing since in many applications it is desirable to remove dependencies with minimal additional adjustments.

We reformulate Theorem E.3 to also include information about the structure of the global minima, and consequently the target predictions regarding those.

**Theorem E.2.** *Let* $c := \lambda_{\mathbf{H}}\lambda_{\mathbf{W}}$. *Any global minimum* $(\tilde{\mathbf{H}}, \tilde{\mathbf{W}})$ *of the regularized UFM-loss*

$$\frac{1}{2M}||\mathbf{W}\mathbf{H} - \mathbf{Y}^{ZCA}||_F^2 + \frac{\lambda_{\mathbf{H}}}{2M}||\mathbf{H}||_F^2 + \frac{\lambda_{\mathbf{W}}}{2}||\mathbf{W}||_F^2, \tag{15}$$

*where* $\lambda_{\mathbf{H}}$ *and* $\lambda_{\mathbf{W}}$ *are non-negative regularization parameters, takes the following form.*

*If $0 < c < 1$, then for any semi-orthogonal matrix $\mathbf{R}$,*

$$\tilde{\mathbf{W}} = \left(\frac{\lambda_{\mathbf{H}}}{\lambda_{\mathbf{W}}}\right)^{1/4} \tilde{\mathbf{A}}^{1/2}\mathbf{R}, \quad \tilde{\mathbf{H}} = \sqrt{\frac{\lambda_{\mathbf{W}}}{\lambda_{\mathbf{H}}}} \tilde{\mathbf{W}}^T \mathbf{Y}^{ZCA}, \tag{16}$$

$$\tilde{\mathbf{W}}\tilde{\mathbf{H}} = (1 - \sqrt{c})\mathbf{Y}^{ZCA},$$

*where $\tilde{\mathbf{A}} = (1 - \sqrt{c})\mathbf{I}_n$.*

*If $c > 1$, then $(\tilde{\mathbf{H}}, \tilde{\mathbf{W}}) = (\mathbf{0}, \mathbf{0})$.*

*Furthermore,*

$$\text{MSE(de-whiten)} = \begin{cases} c\displaystyle\sum_{i=1}^{n} \lambda_i, & \text{if } c < 1, \\ \displaystyle\sum_{i=1}^{n} \lambda_i, & \text{if } c \geq 1, \end{cases}$$

*where $\lambda_i$ is the $i$-th eigenvalue (in descending order) of the original sample covariance matrix $\mathbf{\Sigma}$.*

Before giving the proof, let us first discuss the nature of $\tilde{\mathbf{W}}$ and $\tilde{\mathbf{H}}$ when whitening is applied. In light of (16), properties that hold are:

1. The rows of $\tilde{\mathbf{W}}$ are orthogonal (due to $\tilde{\mathbf{A}} = (1 - \sqrt{c})\mathbf{I}_n$ being diagonal).

2. The rows of $\tilde{\mathbf{W}}$ are equinorm. More specifically,

$$||\tilde{\mathbf{w}}_j||_2^2 = \lambda_{\mathbf{H}}\left(\frac{1}{\sqrt{c}} - 1\right), \quad j = 1, ..., n.$$

3. The angles between the columns of $\tilde{\mathbf{H}}$ are equal to angles between the whitened $\tilde{\mathbf{y}}_i$'s, i.e.,

$$\tilde{\mathbf{H}}^T\tilde{\mathbf{H}} = \lambda_{\mathbf{W}}\left(\frac{1}{\sqrt{c}} - 1\right)(\mathbf{Y}^{ZCA})^T\mathbf{Y}^{ZCA}.$$

**Theorem E.3.**

$$\text{MSE(de-whiten)} = \min\{c, 1\}\sum_{i=1}^{n} \lambda_i. \tag{17}$$

*Proof of Theorem E.3.* It is easily seen that $M^{-1}\mathbf{Y}^{ZCA}(\mathbf{Y}^{ZCA})^T = \mathbf{I}_n$. Thus, the covariance matrix for the whitened targets is the $n \times n$ identity matrix.

**Case $c < 1$:**

By (Andriopoulos et al., 2024)[Theorem 4.1], the optimal predictions after whitening are $\tilde{\mathbf{W}}\tilde{\mathbf{H}} = (1 - \sqrt{c})\mathbf{Y}^{ZCA}$. Our final de-whitened predictions satisfy

$$\hat{\mathbf{Y}} = [\mathbf{\Sigma}^{1/2}]\tilde{\mathbf{W}}\tilde{\mathbf{H}} + \bar{\mathbf{Y}} = (1 - \sqrt{c})(\mathbf{Y} - \bar{\mathbf{Y}}) + \bar{\mathbf{Y}}.$$

Therefore, $\hat{\mathbf{Y}} - \mathbf{Y} = -\sqrt{c}(\mathbf{Y} - \bar{\mathbf{Y}})$, and

$$\frac{(\hat{\mathbf{Y}} - \mathbf{Y})(\hat{\mathbf{Y}} - \mathbf{Y})^T}{M} = c\frac{(\mathbf{Y} - \bar{\mathbf{Y}})(\mathbf{Y} - \bar{\mathbf{Y}})^T}{M} = c\mathbf{\Sigma}.$$

The result for MSE(de-whiten) readily follows by taking traces in both sides.

**Case $c \geq 1$:**

By (Andriopoulos et al., 2024)[Theorem 4.1], the only global minimum is $(\tilde{\mathbf{H}}, \tilde{\mathbf{W}}) = (\mathbf{0}, \mathbf{0})$. Therefore, $\hat{\mathbf{Y}} = \bar{\mathbf{Y}}$, and

$$\frac{(\hat{\mathbf{Y}} - \mathbf{Y})(\hat{\mathbf{Y}} - \mathbf{Y})^T}{M} = \frac{(\mathbf{Y} - \bar{\mathbf{Y}})(\mathbf{Y} - \bar{\mathbf{Y}})^T}{M} = \mathbf{\Sigma}.$$

The result for MSE(de-whiten) readily follows by taking traces in both sides. $\square$

*Proof of Theorem 3.3.* Suppose $0 < c \leq 1$. By Theorem 3.1 and Theorem E.3,

$$
\begin{aligned}
\text{MSE(de-whiten)} - \text{MSE(multi)} &= c \sum_{i=1}^{n} \lambda_i - j^* c - \sum_{i=j^*+1}^{n} \lambda_i \\
&= c \sum_{i=1}^{j^*} \lambda_i + c \sum_{i=j^*+1}^{n} \lambda_i - j^* c - \sum_{i=j^*+1}^{n} \lambda_i \\
&= c \left[ \sum_{i=1}^{j^*} \lambda_i - j^* \right] + (c-1) \sum_{i=j^*+1}^{n} \lambda_i < 0
\end{aligned}
$$

if and only if

$$
\sum_{i=1}^{j^*} \lambda_i - j^* < c^{-1}(1-c) \sum_{i=j^*+1}^{n} \lambda_i.
$$

In the special case when $c < \lambda_{\min}$, i.e., $j^* = n$, the condition reduces to $\bar{\lambda} < 1$.

Suppose $c > 1$. By Theorem 3.1 and Theorem E.3,

$$
\begin{aligned}
\text{MSE(de-whiten)} - \text{MSE(multi)} &= \sum_{i=1}^{n} \lambda_i - j^* c - \sum_{i=j^*+1}^{n} \lambda_i \\
&= \sum_{i=1}^{j^*} \lambda_i + \sum_{i=j^*+1}^{n} \lambda_i - j^* c - \sum_{i=j^*+1}^{n} \lambda_i \\
&= \sum_{i=1}^{j^*} (\lambda_i - c) \geq 0.
\end{aligned}
$$

since by definition $\lambda_i \geq c$, for all $i \leq j^*$. $\qquad\square$

**Theorem E.4.**

$$
\text{MSE(de-normalize)} = \begin{cases} c \sum_{i=1}^{n} \lambda_i, & \text{if } 0 < c < \tilde{\lambda}_{\min}, \\ \sum_{i=1}^{n} \lambda_i, & \text{if } c > \tilde{\lambda}_{\max}. \end{cases} \tag{18}
$$

*Proof of Theorem E.4.* Using the decomposition of $\boldsymbol{\Sigma} = \mathbf{V}^{1/2} \mathbf{P} \mathbf{V}^{1/2}$, it is readily deduced that $M^{-1} \mathbf{Y}^{nrm} (\mathbf{Y}^{nrm})^T = \mathbf{V}^{-\frac{1}{2}} \boldsymbol{\Sigma} \mathbf{V}^{-\frac{1}{2}} = \mathbf{P}$.

**Case** $0 < c < \tilde{\lambda}_{\min}$**:**

By (Andriopoulos et al., 2024)[Theorem 4.1], the optimal predictions after normalization (but before de-normalizing) are $\bar{\mathbf{W}}\bar{\mathbf{H}} = [\mathbf{I}_n - \sqrt{c}\mathbf{P}^{-1/2}]\mathbf{Y}^{nrm}$. Our final de-normalized predictions satisfy

$$
\begin{aligned}
\check{\mathbf{Y}} &= [\mathbf{V}^{1/2}]\bar{\mathbf{W}}\bar{\mathbf{H}} + \bar{\mathbf{Y}} = \mathbf{V}^{1/2}[\mathbf{V}^{-1/2} - \sqrt{c}\mathbf{P}^{-1/2}\mathbf{V}^{-1/2}](\mathbf{Y} - \bar{\mathbf{Y}}) + \bar{\mathbf{Y}} \\
&= \mathbf{Y} - \sqrt{c}\mathbf{V}^{1/2}\mathbf{P}^{-1/2}\mathbf{V}^{-1/2}(\mathbf{Y} - \bar{\mathbf{Y}})
\end{aligned}
$$

Therefore, $\check{\mathbf{Y}} - \mathbf{Y} = -\sqrt{c}\mathbf{V}^{1/2}\mathbf{P}^{-1/2}\mathbf{V}^{-1/2}(\mathbf{Y} - \bar{\mathbf{Y}})$, and

$$
\begin{aligned}
\frac{(\check{\mathbf{Y}} - \mathbf{Y})(\check{\mathbf{Y}} - \mathbf{Y})^T}{M} &= c\mathbf{V}^{1/2}\mathbf{P}^{-1/2} \left[ \mathbf{V}^{-1/2}\boldsymbol{\Sigma}\mathbf{V}^{-1/2} \right] \mathbf{P}^{-1/2}\mathbf{V}^{1/2} \\
&= c\mathbf{V}^{1/2}\mathbf{P}^{-1/2}\mathbf{P}\mathbf{P}^{-1/2}\mathbf{V}^{1/2} = c\mathbf{V}
\end{aligned}
$$

The result for MSE(de-normalize) readily follows by taking traces in both sides.

**Case** $c > \tilde{\lambda}_{\max}$**:**

By (Andriopoulos et al., 2024)[Theorem 4.1], the only global minimum is $(\bar{\mathbf{H}}, \bar{\mathbf{W}}) = (\mathbf{0}, \mathbf{0})$. Therefore, $\check{\mathbf{Y}} = \bar{\mathbf{Y}}$, and

$$\frac{(\check{\mathbf{Y}} - \mathbf{Y})(\check{\mathbf{Y}} - \mathbf{Y})^T}{M} = \frac{(\mathbf{Y} - \bar{\mathbf{Y}})(\mathbf{Y} - \bar{\mathbf{Y}})^T}{M} = \boldsymbol{\Sigma}.$$

The result for MSE(de-normalize) readily follows by taking traces in both sides. □

Let us now examine how MSE(de-normalize) compares to MSE(de-whiten) and to the training MSE with the unprocessed targets. The next theorem follows directly from Theorems E.3-E.4.

**Theorem E.5.** $(i)$ *Suppose* $0 < c < \min\{\lambda_{\min}, \tilde{\lambda}_{\min}\}$. *Then,*

$$MSE(\textit{de-normalize}) = MSE(\textit{de-whiten}).$$

*Furthermore, if* $\bar{\lambda} < 1$, *then* $MSE(\textit{de-normalize}) < MSE$, *where MSE is the training MSE using unprocessed targets. If* $\bar{\lambda} > 1$, *then* $MSE(\textit{de-normalize}) > MSE$. *If* $\bar{\lambda} = 1$, *then* $MSE(\textit{de-normalize}) = MSE$.

*(ii) Suppose* $c > \tilde{\lambda}_{\max}$. *Then,*

$$MSE(\textit{de-normalize}) = MSE(\textit{de-whiten}).$$

*Furthermore,* $MSE(\textit{de-normalize}) \geq MSE$.

Theorem E.5 provides important qualitative insights into the target normalization approach. In practical scenarios, such as with all of our 4 training datasets, we observe that $\lambda_{\min} < 1$ ($\tilde{\lambda}_{\min} \leq 1$ for any arbitrary correlation matrix), as indicated in Table 1. Additionally, test error is typically minimized with small weight decay values, corresponding to $c < 1$. Under these conditions, MSE(de-normalize) = MSE(de-whiten), implying that under the UFM approximation, one can use either target whitening or normalization. As in Theorem 3.3, Theorem E.5 also highlights the crucial role of $\bar{\lambda}$ in determining whether normalization improves or worsens training performance.

## F    FEATURE RELATIONSHIPS ACROSS METHODS

Since the UFM was first proposed to understand feature learning, and especially the neural collapse phenomenon in classification (Papyan et al., 2020) and neural multivariate regression (Andriopoulos et al., 2024), it can also provide insights on the feature $\mathbf{H}$ learned by different methods. For example, how do the features learned by training with original targets or by whitening, and normalization, compare with each other?

Regarding this insightful question, our analysis in the Appendix (Theorem E.2 and the remarks before its proof, and Theorem E.4) explicitly discuss the nature of the optimal $\mathbf{H}_*$ when whitening and normalization are applied respectively. Let us collect the globally optimal learned features and compare them across methods here as well.

**Training with original targets** $\mathbf{Y}$:

$$\mathbf{H}_* = \left(\frac{\lambda_{\mathbf{W}}}{\lambda_{\mathbf{H}}}\right)^{1/4} \mathbf{R}^T [\boldsymbol{\Sigma}^{1/2} - \sqrt{c}\mathbf{I}_n]^{1/2} \mathbf{Y}^{ZCA},$$

where $\mathbf{R} \in \mathbb{R}^{n \times d}$ is semi-orthogonal and $\mathbf{Y}^{ZCA} = \boldsymbol{\Sigma}^{-1/2}(\mathbf{Y} - \bar{\mathbf{Y}})$.

The key observation is that the optimal learned features are formed by two procedures. The term $\boldsymbol{\Sigma}^{1/2} - \sqrt{c}\mathbf{I}_n$ adjusts the covariance of the original target data $\mathbf{Y}$. By subtracting $\sqrt{c}\mathbf{I}_n$ from $\boldsymbol{\Sigma}^{1/2}$, small eigenvalues are regularized or "shrunk", effectively denoising the original target data. Taking the root of the result further scales the eigenvalues nonlinearly, emphasizing stronger signal directions. The first procedure consists of the whitened target data undergoing adaptive scaling, i.e., the multiplication $[\boldsymbol{\Sigma}^{1/2} - \sqrt{c}\mathbf{I}_n]^{1/2} \mathbf{Y}^{ZCA}$ re-weights the whitened target data using the thresholded eigenvalues from $\boldsymbol{\Sigma}$. Directions aligned with strong original covariance are amplified, while weak/noisy directions are zeroed out. The second procedure consists of a rotation of the first procedure's outcome. The semi-orthogonal matrix $\mathbf{R}^T$ acts as a rotation, redistributing the features into a coordinate system that preserves distances but may optimize for properties like orthogonality or sparsity. To summarize,

the two procedures ensure a denoised, lower dimensional feature representation that retains only statistically significant components from the covariance of the original target data.

For single-task $i$:

$$\mathbf{H}_*^{(i)} = \left(\frac{\lambda_{\mathbf{W}}}{\lambda_{\mathbf{H}}}\right)^{1/4} \mathbf{R}^T[\sigma_i - \sqrt{c}]^{1/2}\sigma_i^{-1}\mathbf{y}^{(i)},$$

where $\mathbf{y}^{(i)}$ is the $i$-th row of $\mathbf{Y}$. There is no a priori relationship between $\mathbf{H}_*$ and $\mathbf{H}_*^{(i)}$. However, we note that in the special case when the targets are uncorrelated, it is easy to see that $\mathbf{H}_*^{(i)}$ is the $i$-th row of $\mathbf{H}_*$.

**Training with whitened targets $\mathbf{Y}^{ZCA}$:**

$$\mathbf{H}_* = \left(\frac{\lambda_{\mathbf{W}}}{\lambda_{\mathbf{H}}}\right)^{1/4} \mathbf{R}^T[\mathbf{I}_n^{1/2} - \sqrt{c}\mathbf{I}_n]^{1/2}\mathbf{Y}^{ZCA}.$$

In this case, the form of the optimal learned features is retained with the difference that the $n \times n$ identity matrix $\mathbf{I}_n$ takes the place of $\boldsymbol{\Sigma}$. The whitened target data is simply scaled down by $\sqrt{1 - \sqrt{c}}$, if $c < 1$, akin to mild regularization. No denoising (eigenvalue thresholding) takes place, and all the directions are kept.

**Training with normalized targets $\mathbf{Y}^{nrm} = \mathbf{V}^{-1/2}(\mathbf{Y} - \bar{\mathbf{Y}})$:**

Recall that we have used $\mathbf{P}$ to denote the correlation matrix of the targets. Then,

$$\mathbf{H}_* = \left(\frac{\lambda_{\mathbf{W}}}{\lambda_{\mathbf{H}}}\right)^{1/4} \mathbf{R}^T[\mathbf{P}^{1/2} - \sqrt{c}\mathbf{I}_n]^{1/2}\mathbf{Y}^{ZCA-cor},$$

where $\mathbf{Y}^{ZCA-cor} := \mathbf{P}^{-1/2}\mathbf{Y}^{nrm}$ is referred to in the literature as the ZCA-cor whitening (Kessy et al., 2018), and it is the unique whitening procedure that makes the transformed normalized targets as similar as possible to the original normalized targets, the similarity being in terms of the cross-correlation between the former and the latter.

The optimal leaned features are obtained in accordance to the analysis that we have outlined when training with original targets. Here, the role of $\boldsymbol{\Sigma}$ is played by $\mathbf{P}$ and in the place of $\mathbf{Y}^{ZCA}$, we have $\mathbf{Y}^{ZCA-cor}$.

## G  GENERALIZATION BOUNDS

Let $\mathcal{X}$ denote the input space and $\mathcal{Y}$ the target space, which regarding the learning problem of neural multivariate regression, is a subset of $\mathbb{R}^n$. Here, we adopt the stochastic scenario and will denote by $\mathcal{D}$ a distribution over $\mathcal{X} \times \mathcal{Y}$. In the supervised learning scenario, the learner receives training examples $S := \{(\mathbf{x}_i, \mathbf{y}_i), i = 1, ..., M\} \in (\mathcal{X} \times \mathcal{Y})^M$ drawn in a i.i.d. manner according to $\mathcal{D}$. The deterministic scenario where input points admit a unique target value determined by a target function $f : \mathcal{X} \to \mathcal{Y}$ is a straightforward special case.

We denote by $L : \mathcal{Y} \times \mathcal{Y} \to \mathbb{R}_+$ the loss function used to measure the magnitude of the difference between the vector-valued target predicted and the "true" or "correct" one. The most common loss function used in neural multivariate regression is the $L_p$-loss defined by $L(\mathbf{y}, \mathbf{y}') = ||\mathbf{y} - \mathbf{y}'||_p$, for some $p \geq 1$, and every $\mathbf{y}, \mathbf{y}' \in \mathcal{Y}$.

Given a hypothesis set $\mathcal{H}$, that is a set which contains all the functions mapping the input space $\mathcal{X}$ to the target space $\mathcal{Y}$, neural multivariate regression tasks consist of using a set of training examples $S$ to find a hypothesis $h \in \mathcal{H}$ with small generalization error $R(h)$ with respect to the "true" or "correct" *target function* mapping inputs to targets in $S$.

**Definition G.1** (Generalization error). Given a hypothesis $h \in \mathcal{H}$, a *target function*, and an underlying distribution $\mathcal{D}$, the generalization error is defined by

$$R(h) := \mathbb{E}_{(\mathbf{x},\mathbf{y})\sim\mathcal{D}} \left[L(h(\mathbf{x}), \mathbf{y})\right],$$

where $L : \mathcal{Y} \times \mathcal{Y} \to \mathbb{R}_+$ is the loss function used to measure the magnitude of error.

The generalization error is not directly accessible to the learner since both the distribution $\mathcal{D}$ and the *target function* are unknown. However, the learner can measure the empirical error of a hypothesis on a set of training examples $S$.

**Definition G.2** (Training error). Given a hypothesis $h \in \mathcal{H}$, a *target function*, and a training set $S$, the training error is

$$\hat{R}_S(h) := \frac{1}{M} \sum_{i=1}^{M} L(h(\mathbf{x}_i), \mathbf{y}_i).$$

Thus, the training error of $h$ is its average error over the training set $S$, while the generalization error is its expected error based on the distribution $\mathcal{D}$. If $L$ is the squared loss, the training error represents the training MSE of $h$ on $S$.

The theoretical results presented below are based on the assumption that the neural multivariate regression problem is bounded, that is when the loss function is bounded above by some $K > 0$, i.e., $L(\mathbf{y}, \mathbf{y}') \le K$, for all $\mathbf{y}, \mathbf{y}' \in \mathcal{Y}$, or, more strictly, when $L(h(\mathbf{x}), \mathbf{y}) \le K$, for all $h \in \mathcal{H}$, and $(\mathbf{x}, \mathbf{y}) \in \mathcal{X} \times \mathcal{Y}$.

## G.1 Generalization bounds for kernel-based hypotheses

Generalization bounds based on Rademacher complexity (RC), a term that captures the richness of a family of functions by measuring the degree to which a hypothesis set can fit random noise, were presented in (Mohri, 2018), e.g., Theorem 11.3 therein. These generalization bounds suggest a trade-off between reducing the training MSE and controlling the RC of the hypothesis set. A richer or more complex hypothesis set achieves a small training MSE but has high RC, while a poorer or more simple hypothesis set has small RC but achieves high training MSE. The aim is to control this trade-off. An important benefit of the learning bounds in Mohri (2018)[Theorem 11.3] is that they are data dependent. This can lead to more accurate learning guarantees. For kernel-based hypotheses upper bounds on the RC can be used directly to derive generalization bounds depending on the trace of the kernel matrix or the maximum diagonal entry.

**Definition G.3** (Kernel). A function $K : \mathcal{X} \times \mathcal{X} \to \mathbb{R}$ is said to be a positive definite symmetric (PDS) kernel if for any $\{\mathbf{x}_i : i = 1, ..., M\} \in \mathcal{X}$, the matrix $K := [K(\mathbf{x}_i, \mathbf{x}_j)]_{i,j} \in \mathbb{R}^{M \times M}$ is symmetric positive semi-definite (SPSD).

For a PDS kernel $K : \mathcal{X} \times \mathcal{X} \to \mathbb{R}$, there exists a Hilbert space $\mathbb{H}$ and a mapping $\Phi : \mathcal{X} \to \mathbb{H}$ such that:

$$K(\mathbf{x}, \mathbf{x}') = \langle \Phi(\mathbf{x}), \Phi(\mathbf{x}') \rangle, \qquad \forall \mathbf{x}, \mathbf{x}' \in \mathcal{X}.$$

For a proof of this result, we refer the reader to Mohri (2018)[Theorem 6.8].

A generalization bound for (univariate) linear regression with bounded linear hypotheses in a feature space defined by a PDS kernel was presented in Mohri (2018)[Theorem 11.11]. For simplicity, we give the generalization bound for the squared loss.

**Theorem G.4.** *Let $K : \mathcal{X} \times \mathcal{X} \to \mathbb{R}$ be a PDS kernel, $\Phi : \mathcal{X} \to \mathbb{H}$ be a feature mapping associated with $K$, and*

$$\mathcal{H} := \{\mathbf{x} \mapsto \mathbf{w} \cdot \Phi(\mathbf{x}) : ||\mathbf{w}|| \le \Lambda_{\mathbf{w}}\}$$

*be the family of bounded linear hypotheses corresponding to the optimization problem*

$$\min_{\mathbf{w}} \frac{1}{M} \sum_{i=1}^{M} (\mathbf{w} \cdot \Phi(\mathbf{x}_i) - \mathbf{y}_i)^2, \text{ subject to } ||\mathbf{w}|| \le \Lambda_{\mathbf{w}},$$

*for a training set $S = \{(\mathbf{x}_i, \mathbf{y}_i) : i = 1, ..., M\}$.*

- *Assume that there exists $K > 0$ such that $|h(\mathbf{x}) - \mathbf{y}| \le K$, for all $(\mathbf{x}, \mathbf{y}) \in \mathcal{X} \times \mathcal{Y}$.*

- *Let $tr\left([K(\mathbf{x}_i, \mathbf{x}_j)]_{i,j}\right) \le Mr^2$, for any training set $S$ of size $M$.*

*Then, for any $\delta > 0$, with probability at least $1 - \delta$, the following inequality holds for all $h \in \mathcal{H}$:*

$$\mathbb{E}_{(\mathbf{x}, \mathbf{y}) \in \mathcal{D}}[|h(\mathbf{x}) - \mathbf{y}|^2] \le \text{MSE}_S(h) + 4K \frac{r \Lambda_{\mathbf{w}}}{\sqrt{M}} + 3K^2 \sqrt{\frac{\log \frac{2}{\delta}}{2M}}. \tag{19}$$

The generalization bound of the theorem above suggest minimizing a trade-off between the training MSE, denoted in (19) by $\text{MSE}_S(h)$, and the norm of the weight vector $\mathbf{w}$. The third term adds an error dependent on the confidence level $\delta$ and the size of the training set $M$.

## G.2 APPLICATION TO THE UFM VIA THE LAYER-PEELED MODEL

Another formulation of the UFM is the so-called Layer-Peeled Model introduced by Fang et al. (2021) as:

$$\min_{\mathbf{W},\mathbf{H}} ||\mathbf{W}\mathbf{H} - \mathbf{Y}||_F^2, \text{ subject to } \begin{cases} ||\mathbf{W}||_F & \leq \Lambda_\mathbf{W}, \\[2ex] \dfrac{||\mathbf{H}||_F}{\sqrt{M}} & \leq \Lambda_\mathbf{H}, \end{cases} \tag{20}$$

where $\mathbf{W} \in \mathbb{R}^{n \times d}$ is, as in (1), a linear classifier in the last layer, $\mathbf{H} = [\mathbf{h}_1, ..., \mathbf{h}_M] \in \mathbb{R}^{d \times M}$, where $\mathbf{h}_i := \mathbf{h}_\theta(\mathbf{x}_i)$ is the $d$-dimensional last-layer activation/feature of the $i$-th training sample, and $\Lambda_\mathbf{W}, \Lambda_\mathbf{H}$ are positive scalars constraining the matrix-norms of $\mathbf{W}$ and $\mathbf{H}$ respectively.

A constrained minimization problem can be solved by means of the Karush-Kuhn-Tucker (KKT) multiplier method, which minimizes a function subject to inequality constraints. The KKT multiplier method states that, under some regularity conditions (all met here), there exist constants $\nu_\mathbf{W}, \nu_\mathbf{H} \geq 0$, called the multipliers, such that the solution $(\mathbf{W}(\nu_\mathbf{W}), \mathbf{H}(\nu_\mathbf{H}))$ of the constrained minimization problem (20) satisfies the so-called KKT conditions.

- The first condition (referred to as the stationarity condition) demands that the gradients of the Langrangian

$$\Delta(\mathbf{W}, \mathbf{H}) := ||\mathbf{W}\mathbf{H} - \mathbf{Y}||_F^2 + \nu_\mathbf{W}(||\mathbf{W}||_F^2 - \Lambda_\mathbf{W}^2) + \nu_\mathbf{H}(||\mathbf{H}||_F^2 - M\Lambda_\mathbf{H}^2),$$

associated with the minimization problem (1), i.e., the UFM, is 0 at the solution $(\mathbf{W}(\nu_\mathbf{W}), \mathbf{H}(\nu_\mathbf{H}))$. More specifically,

$$\left.\frac{\partial \Delta}{\partial \mathbf{W}}\right|_{(\mathbf{W}(\nu_\mathbf{W}), \mathbf{H}(\nu_\mathbf{H}))} = 2(\mathbf{W}(\nu_\mathbf{W})\mathbf{H}(\nu_\mathbf{H}) - \mathbf{Y})\mathbf{H}(\nu_\mathbf{H})^T + 2\nu_\mathbf{W}\mathbf{W}(\nu_\mathbf{W}), \tag{21}$$

$$\left.\frac{\partial \Delta}{\partial \mathbf{H}}\right|_{(\mathbf{W}(\nu_\mathbf{W}), \mathbf{H}(\nu_\mathbf{H}))} = 2\mathbf{W}(\nu_\mathbf{W})^T(\mathbf{W}(\nu_\mathbf{W})\mathbf{H}(\nu_\mathbf{H}) - \mathbf{Y}) + 2\nu_\mathbf{H}\mathbf{H}(\nu_\mathbf{H}). \tag{22}$$

- The second KKT condition (referred to as the complimentarity condition) requires that

$$\nu_\mathbf{W}(||\mathbf{W}(\nu_\mathbf{W})||_F^2 - \Lambda_\mathbf{W}^2) = 0, \qquad \nu_\mathbf{H}(||\mathbf{H}(\nu_\mathbf{H})||_F^2 - M\Lambda_\mathbf{H}^2) = 0. \tag{23}$$

If

$$\nu_\mathbf{W} = \lambda_\mathbf{W}, \qquad \nu_\mathbf{H} = \lambda_\mathbf{H},$$

$$\Lambda_\mathbf{W} = ||\mathbf{W}(\lambda_\mathbf{W})||_F, \qquad \Lambda_\mathbf{H} = \frac{||\mathbf{H}(\lambda_\mathbf{H})||_F}{\sqrt{M}},$$

the UFM solution $(\mathbf{W}(\lambda_\mathbf{W}), \mathbf{H}(\lambda_\mathbf{H}))$ satisfies (21)-(23). Therefore, the theorem that follows is immediately deduced.

**Theorem G.5.** *If*

$$\Lambda_\mathbf{W} = ||\mathbf{W}(\lambda_\mathbf{W})||_F, \qquad \Lambda_\mathbf{H} = \frac{||\mathbf{H}(\lambda_\mathbf{H})||_F}{\sqrt{M}},$$

*the minimization problems of the UFM (1) and the Layer-Peeled Model (20) have the same solution.*

*Remark* G.6. The UFM solutions $\mathbf{W}(\lambda_\mathbf{W})$ and $\mathbf{H}(\lambda_\mathbf{H})$ are always to be found on the boundary of the Layer-Peeled Model constraints, parameterized by $\{(\mathbf{W}, \mathbf{H}) : ||\mathbf{W}||_F \leq \Lambda_\mathbf{W}, ||\mathbf{H}||_F \leq M\Lambda_\mathbf{H}\}$ for some $\Lambda_\mathbf{W}, \Lambda_\mathbf{H} > 0$. The size of the spherical constraints of the Layer-Peeled Model shrink as the regularizing constants of the UFM increase, and eventually in the $\lambda_\mathbf{W}, \lambda_\mathbf{H} \to \infty$-limit, $\Lambda_\mathbf{W}, \Lambda_\mathbf{H} \to 0$. This follows from the closed-form functions of the minimizers for the UFM with respect to each other, i.e.,

$$\lim_{\lambda_\mathbf{w} \to \infty} \mathbf{W}(\lambda_\mathbf{W}) = \lim_{\lambda_\mathbf{W} \to \infty} \mathbf{Y}\mathbf{H}^T[\mathbf{H}\mathbf{H}^T + M\lambda_\mathbf{W}]^{-1} = 0,$$

$$\lim_{\lambda_\mathbf{H} \to \infty} \mathbf{H}(\lambda_\mathbf{H}) = \lim_{\lambda_\mathbf{H} \to \infty} \mathbf{W}^T[\mathbf{W}\mathbf{W}^T + \lambda_\mathbf{H}\mathbf{I}_n]^{-1}\mathbf{Y} = 0.$$

*Remark* G.7. Suppose $c < \lambda_{\min}$. By applying Andriopoulos et al. (2024)[Corollary 4.2 (ii)-(iii)] to the $n$-dimensional case, the following relation holds:

$$\Lambda_{\mathbf{W}} = ||\mathbf{W}(\lambda_{\mathbf{W}})||_F = \left(\frac{\lambda_{\mathbf{H}}}{\lambda_{\mathbf{W}}}\right)^{1/4} \left[\operatorname{tr}\left(\mathbf{\Sigma}^{1/2} - \sqrt{c}\mathbf{I}_n\right)\right]^{1/2},$$

$$\Lambda_{\mathbf{H}} = \frac{||\mathbf{H}(\lambda_{\mathbf{H}})||_F}{\sqrt{M}} = \sqrt{\frac{\lambda_{\mathbf{W}}}{\lambda_{\mathbf{H}}}}||\mathbf{W}(\lambda_{\mathbf{W}})||_F = \sqrt{\frac{\lambda_{\mathbf{W}}}{\lambda_{\mathbf{H}}}}\Lambda_{\mathbf{W}}.$$

Recall that $\mathbf{h}_i := \mathbf{h}_\theta(x_i) \in \mathbb{R}^d$ for all $i = 1, ..., M$, where $\mathbf{h}_\theta$ is associated with the PDS kernel $K = \mathbf{H}^T\mathbf{H}$, i.e., $[\mathbf{h}_i \cdot \mathbf{h}_j]_{i,j} \in \mathbb{R}^{M \times M}$ is the covariance matrix of the $\mathbf{h}_i$'s and as such it is symmetric and positive semi-definite. Under the constraints that $\mathbf{W}$ and $\mathbf{H}$ are subject to:

$$\operatorname{tr}(K) = \operatorname{tr}(\mathbf{H}^T\mathbf{H}) = ||\mathbf{H}||_F^2 \leq M\Lambda_{\mathbf{H}}^2.$$

Under the UFM, when $c < \lambda_{\min}$, $\text{MSE}_S(h) = c = \lambda_{\mathbf{W}}\lambda_{\mathbf{H}}$ for any labeled sample $S$, see Theorem 3.1. Combining the points above, the generalization bound of (19) yields

$$\mathbb{E}_{(\mathbf{x},\mathbf{y}) \in \mathcal{D}}[|h(\mathbf{x}) - \mathbf{y}|^2] \leq nc + \mathcal{O}\left(\frac{C}{\sqrt{M}}\right) + \mathcal{O}\left(\sqrt{\frac{\log 2\delta^{-1}}{M}}\right),$$

where $C := \Lambda_{\mathbf{W}}\Lambda_{\mathbf{H}}$. In the special case in which we set $\lambda_{\mathbf{W}} = \lambda_{\mathbf{H}}$, we have $c = \lambda_{\mathbf{W}}^2$, and $C = \Lambda_{\mathbf{W}}^2 = \operatorname{tr}\left(\mathbf{\Sigma}^{1/2} - \lambda_{\mathbf{W}}\mathbf{I}_n\right)$, and the generalization bound reads

$$\mathbb{E}_{(\mathbf{x},\mathbf{y}) \in \mathcal{D}}[|h(\mathbf{x}) - \mathbf{y}|^2] \leq n\lambda_{\mathbf{W}}^2 + \mathcal{O}\left(\frac{\operatorname{tr}\left(\mathbf{\Sigma}^{1/2} - \lambda_{\mathbf{W}}\mathbf{I}_n\right)}{\sqrt{M}}\right) + R, \tag{24}$$

where, for a fixed confidence level $\delta \in (0, 1)$, $\lim_{M \to \infty} R = 0$.

For single task $i$, the right-hand side (wit the remainder term) of (24) becomes

$$\text{GB}^{(i)} := \lambda_{\mathbf{W}}^2 + \mathcal{O}(M^{-1/2}(\sigma - \lambda_{\mathbf{W}})) + R.$$

Using this, we can directly derive an upper bound for the generalization error of the $n$-single tasks neural regression problem:

$$\text{GB(n-single)} \leq \sum_{i=1}^{n} \text{GB}^{(i)} = n\lambda_{\mathbf{W}}^2 + \mathcal{O}\left(\frac{\sum_{i=1}^{n} \sigma_i - n\lambda_{\mathbf{W}}}{\sqrt{M}}\right) + \tilde{R}, \tag{25}$$

where, for a fixed confidence level $\delta \in (0, 1)$, $\lim_{M \to \infty} \tilde{R} = 0$.

Because $\sum_{i=1}^{n} \sigma_i = \sum_{i=1}^{n} \sqrt{\Sigma_{ii}} \geq \operatorname{tr}(\mathbf{\Sigma}^{1/2})$, the $\mathcal{O}$-term in (25) is $\geq$ than the $\mathcal{O}$-term in (24), so the whole right-hand side of the bound in (25) is $\geq$ than the right-hand side of the bound in (25).

If the two test MSEs concentrate near their respective bounds, then the multi-task test MSE is smaller than that of the $n$ single tasks; thus the gain from multi-tasking is tightest when $\mathbf{\Sigma}$ is non-diagonal and the features across tasks are correlated.

