# OpenReview forum: "Neural Multivariate Regression with Multi-Task Learning and Target Preprocessing"
_ICLR.cc/2026/Workshop/Sci4DL — Sci4DL 2026_

### Official Review · Reviewer_fvqb · 2026-02-24

**Fit:** 2
**Significance:** 2
**Confidence:** 1

**Summary:**

Multivariate regression with neural networks is studied using a previously-proposed theoretical model (UFM). The analysis shines a theoretical light on two issues (multitask learning and target whitening), and it in both cases the theoretical predictions are confirmed in a number of numerical experiments.

**Strengths:**

The submission is clearly written, and shows that a very simple model can elucidate some real-world issues.

**Suggestions:**

It is interesting that the weight decay parameter plays such an important role in these findings. If the authors have strong intuitions on why this is the case, I would like to see those in the discussion.

---

### Official Review · Reviewer_Ed4p · 2026-02-27

**Fit:** 3
**Significance:** 3
**Confidence:** 2

**Summary:**

The authors utilize the closed-form Unconstrained Feature Model (UFM) to derive predictions and testable hypotheses for multi-task learning and target whitening. Experiments are conducted across 4 datasets and MLP/ResNet18 models. Multi-task models are found to achieve lower training MSE across different SGD weight decays and to improve generalization. There is little difference in train MSE between whitening and normalizing, and the impact depends on the average eigenvalues. This enables practical insights. Results align with the UFM hypotheses, showing that the UFM model can generate predictions that transfer to empirical results.

**Strengths:**

The paper is well-organized and the experiments/results are clear. The significance of the work is strongly motivated. Empirically validating weight decay levels is a sound basis. I recommend for a contributed talk due to the methodology of using simplified theoretical frameworks to guide empirical results.

**Suggestions:**

More details on the UFM in section 3. The UFM. would be helpful. From my limited understanding of the UFM, the feature maps are only defined over the training set; this is an important point worth noting. The empirical results show that the hypotheses generalize to test data, so it could be worth briefly discussing this gap.

How related are the multiple different tasks for multi-task models? I don't have a lot of knowledge in this area, but would the UFM predict that the multi-task advantage disappears if the tasks are dissimilar?

---

### Meta-Review · Area_Chair_aXk5 · 2026-03-02

**Recommendation:** Accept

**Metareview:**

This work uses a theoretical model (the Unconstrained Feature Model) to derive testable predictions for multi-task learning and target whitening. The methodology is perfectly aligned with the workshop, and its ability to provide practical insights makes the work very interesting.

---

### Decision · Program_Chairs · 2026-03-02

Accept